# Exploratory study on the application of laser-induced cavitation bubble in the treatment of fallopian tube obstruction

**Dongdong Wang**[1], **Fang Yang**[2‡], **Weinan Gao**[1], **Yong Bi**[1*],
**Xuehong Zhu**[3], **Zhong Lin**[3]

**1** Technical Institute of Physics and Chemistry, Chinese Academy of Sciences, Beijing, China,
**2** China-Japan Friendship Hospital, Beijing, China, **3** The Reproductive Hospital of Guangxi Zhuang Autonomous Region, Nanning, China

‡ This author is the cofirst author.

* biyong@mail.ipc.ac.cn

**Data availability statement:** All relevant data are within the manuscript and its Supporting Information files.

## Abstract

**Significance:** Globally, infertility affects 10–15% of couples, with Fallopian Tube Obstruction (FTO) being a principal cause. Current therapeutic options are inadequate, prompting a demand for effective, less invasive treatments with lower recurrence.

**Aim:** The purpose of this study is to explore the feasibility of employing laser-induced cavitation bubbles (LICB) technology for the treatment of FTO and to experimentally ascertain the optimal laser parameters for this approach.

**Method:** Utilizing an Er:YAG pulsed laser ($2.94\,\mu m$, $200\,\mu s$ pulse width, $10\,Hz$), laser energy was transmitted through a $450\,\mu m$ sapphire fiber. The study involved 13 rats categorized into control, model, and experimental groups. The laser, with energies of $8.3\,mJ$, $12.7\,mJ$, and $15.3\,mJ$, was applied through the sapphire fibers. The assessment criteria incorporated Hematoxylin-Eosin (HE) staining of the fallopian tubes across all groups, evaluating tubal patency and wall damage to ascertain the most effective laser parameters.

**Results:** In this experiment, the cumulative recanalization success rate in rat models of FTO treated with LICB was observed to be 75%. Specifically, a recanalization rate of 33.3% was achieved with the application of 8 mJ laser energy. The use of 12.7 mJ laser energy resulted in an increased success rate of 85.7%. However, while the application of 15.3 mJ laser energy achieved a 100% recanalization rate, it was accompanied by the formation of hemorrhagic spots on the fallopian tube walls, indicating thermal damage due to the higher energy levels. Optimal treatment parameters were identified as 12.7 mJ laser energy, 10 Hz frequency, and 10-second application.

**Conclusions:** This research suggests that LICB technology can effectively clear fallopian tube obstructions while causing acceptable levels of damage. This indicates its potential as a valuable treatment method worthy of further research for facilitating tubal recanalization.

**Funding:** The work has been funded by Nanning Scientific Research and Technological Development Plans Project (No. 20233067) and National Natural Science Foundation of China (No. 62205350). The funders had no role in study design, data collection and analysis, decision to publish, or preparation of the manuscript.

**Competing interests:** The authors have declared that no competing interests exist.

## Introduction

The desire to have children is a deeply ingrained human aspiration. Infertility, transcending a mere reproductive health issue, significantly affects the social and psychological well-being of individuals and families [1]. In the context of changing lifestyles, socio-economic factors, and environmental influences, infertility is an escalating challenge, affecting approximately 10% to 15% of married couples globally, as estimated by the World Health Organization (WHO) [2–4]. Couples are generally advised to seek medical evaluation if conception has not occurred after one year of regular, unprotected intercourse [5]. Fallopian tube obstruction (FTO) is a major cause of infertility, accounting for 30% to 40% of such cases [6,7].

Current interventions for tubal-factor infertility primarily include tuboplasty, tubal cannulation, and assisted reproductive technologies (ART) [8–10]. Tubal cannulation under guided laparoscopy, notably for proximal tubal obstructions, has up to an 85% success rate [5]. However, these interventions often result in less favorable outcomes for over 20% of patients, with postoperative conception rates around 30% [11,12]. The high cost, invasiveness, and potential for postoperative re-obstruction are significant deterrents [13,14]. Increasingly, patients with tubal obstruction are turning to ART for treatment. However, this approach not only requires significant time and financial investment but also carries risks such as ovarian hyperstimulation syndrome and multiple pregnancies [15]. Consequently, the pursuit of a less painful, more efficacious, and minimally recurrent method for treating tubal obstruction remains a primary focus in gynecology.

Cavitation, a well-recognized phenomenon in fluid mechanics [16,17], is now being applied beneficially in medical research. Laser-Induced Cavitation Bubble (LICB) technology employs lasers to create plasma or vapor bubbles in liquids, which expand rapidly and collapse under pressure differential, generating shockwaves and microjets [18–23]. This technology, notable for its precision and reproducibility, has diverse medical applications, such as in endodontics for root canal cleaning [24–28], in urolithiasis treatment for stone fragmentation [29], and in thrombolysis for clot ablation [30–32].

LICB technology has demonstrated distinct advantages in medical applications, particularly in ablating lesions within micro-channel structured tissues. Unfortunately, its application in recanalizing FTO is yet to be explored, as current literature lacks reports on such studies [33]. Considering the absence of optimal treatments for FTO, the potential of LICB technology to ablate lesions through laser energy transmitted via optical fibers presents an innovative and promising avenue for investigation in the field of reproductive medicine.

In this research, rat-models of FTO were employed to assess the feasibility of LICB technology for recanalization. An Er:YAG pulsed laser, emitting 2.94 μm wavelength light matching water's absorption peak, was used, reducing potential thermal damage to the fallopian tube walls [34]. Laser energy, transmitted via a sapphire fiber, targeted the obstruction site, triggering cavitation bubbles in the surrounding fluid. The ensuing shockwaves from these bubbles effectively disintegrated the obstructive tissue. Tube patency was evaluated with hydrotubation, and various laser energies were tested to understand their influence on bubble dynamics and recanalization success, thereby identifying optimal laser parameters through comparative appearance and HE staining analyses.

## Materials and methods

### Study protocol

In this research, a 15% phenol paste-induced fallopian tube obstruction model in rats was established to simulate FTO. The phenol paste triggered extensive epithelial injury and

inflammation in the fallopian tubes, likely causing obstruction due to early collagen deposition and hemorrhagic content [35]. The modeling efficacy was verified by the permeation of methylene blue in the uterus to the ovaries. Laser energy parameters were determined based on the cavitation bubble sizes induced in a free fluid medium and the diameter measurements at the uterine horn vicinity in rats. Rats in the model group were segregated into three subgroups, each subjected to varying laser energies for recanalization, following confirmation of successful modeling via methylene blue injection. The results were critically evaluated to ascertain the most effective laser parameters for FTO treatment. The detailed experimental protocol is illustrated in Fig 1.

## Animal protocol

This investigation utilized 14 specific pathogen-free (SPF) female Sprague-Dawley (SD) rats, aged 7 weeks, with body weights ranging from 250 to 300 grams. These rats were procured from Beijing Weishanglide Biotechnology Co., Ltd., holding a qualification certificate number SCXK (Beijing) 2021–0010. The animals were housed under the same facility's certification, SYXK (Beijing) 2021–0056. The environmental conditions of the housing facility were stringently regulated, including a maintained temperature of $(22 \pm 3)°C$, humidity levels between

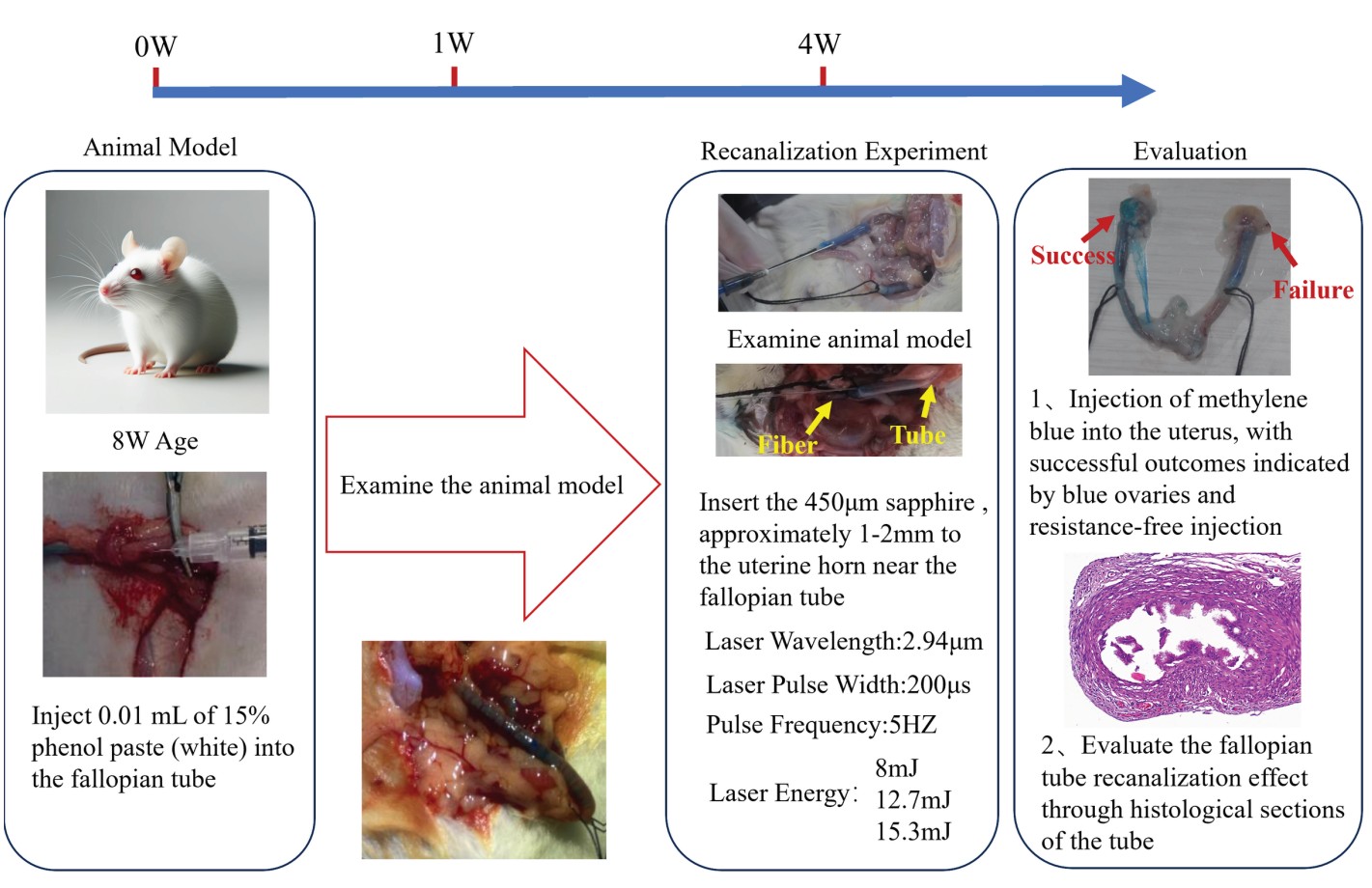

**Fig 1. Experimental protocol of LICB treatment for FTO in rats.**

40-80%, and a consistent 12-hour light/dark cycle. The rats had unrestricted access to food and water throughout the study.

The animal research protocol employed in this study was rigorously reviewed and received approval from the Institutional Animal Care and Use Committee (IACUC) of the Technical Institute of Physics and Chemistry, Chinese Academy of Sciences (CAS). The conduct of all animal experiments adhered to the stringent guidelines and regulations established for animal experimentation by the Technical Institute of Physics and Chemistry, CAS.

Following 7 days of acclimatization feeding, the rats were assigned numbers based on their weight. Two rats were randomly selected to constitute the control group, while the remaining 12 were subjected to the creation of a chronic fallopian tube inflammatory obstruction (FTO) model using the phenol paste method. The rats received general anesthesia via intraperitoneal injection of 10% hydrochloric acid cerazine (5 ml/kg) from Jilin Huamu Animal Health Care Co., Ltd. (veterinary medicine number 070012926). Subsequently, they were immobilized on the operating table, with the abdominal skin prepared, the surgical area sterilized, and draped.

A 1 cm longitudinal incision, positioned approximately 0.5 cm above the pubic symphysis, was made to systematically expose the bilateral uterus, fallopian tubes, and ovaries (Fig 2a). Near the uterine horn, adjacent to the fallopian tube, a needle was inserted for injecting 0.01 ml of 15% phenol paste into both fallopian tubes (Fig 2b).

One week postoperatively, one rat from both the control and model groups was randomly euthanized for model verification: (1) Control group: The fallopian tube lumen appeared regular in shape with elastic, soft walls and a fresh color, free from congestion, swelling, twisting, or other lesions, and without adhesion to adjacent tissues. A needle was inserted at the uterus-fallopian tube junction, and approximately 3 mL of diluted methylene blue solution (Jichuan Pharmaceutical Group Co., Ltd.) was injected towards the fallopian tube-ovary direction. The unobstructed flow of methylene blue and its appearance in the ovaries (Fig 2c) confirmed patency; (2) Model group: The fallopian tube lumen exhibited twisting and deformation, with localized lumen enlargement and extensive adhesion to surrounding tissues. The resistance to methylene blue injection and absence of blue dyeing in the ovaries indicated FTO and successful model establishment (Fig 2d).

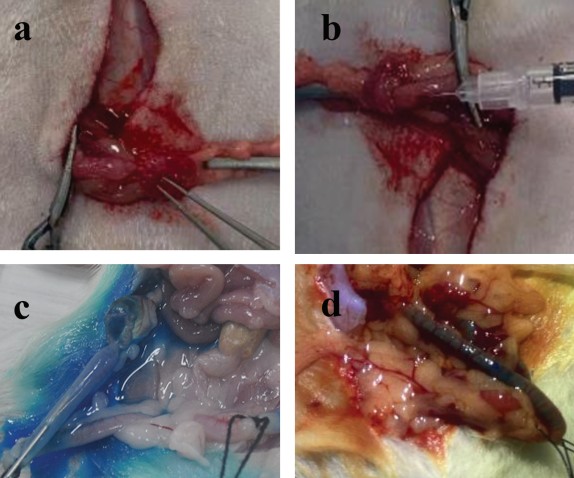

**Fig 2. Process of establishing and testing the rat FTO model.**

We administered an overdose of anesthetic (100 mg/kg of 3% sodium pentobarbital via intraperitoneal injection) to euthanize each rat. This method complies with the AVMA Guidelines for the Euthanasia of Animals to minimize distress.

## Experimental platform

Fig 3 illustrates the design of the laser surgery experimental setup. The apparatus utilizes an Er:YAG pulsed laser, characterized by a wavelength of 2.94 μm and a pulse width of 200 μs, with an adjustable energy range from 0 to 100 mJ. The laser operates at a frequency of 10 Hz. To channel the laser energy to the targeted area, a sapphire fiber (450 μm in diameter and 10 cm in length, sourced from Jingying Optoelectronics, China) is utilized.

The laser activation is managed by square wave pulses generated from a waveform generator (model DG1002U, produced by Rigol, China), which then triggers the output of pulsed lasers. Concurrently, methylene blue solution is injected into the fallopian tube to facilitate the identification of the obstructed areas.

For the surgical procedure, rats are positioned on a specially designed movable surgical platform. This platform enables precise adjustments, allowing the energy output end of the sapphire fiber to be accurately positioned near the uterine corner. Such positioning is critical for the effective removal of obstructive tissue in the fallopian tubes, leveraging the cavitation effect induced by the laser.During surgery, the rats' respiration and body temperature were closely monitored to ensure they remained in a deeply anesthetized and pain-free state.

## In vitro experiment

To establish appropriate laser parameters for the experiment, this study initially conducted in vitro investigations into the generation of Laser-Induced Cavitation Bubbles (LICB) in

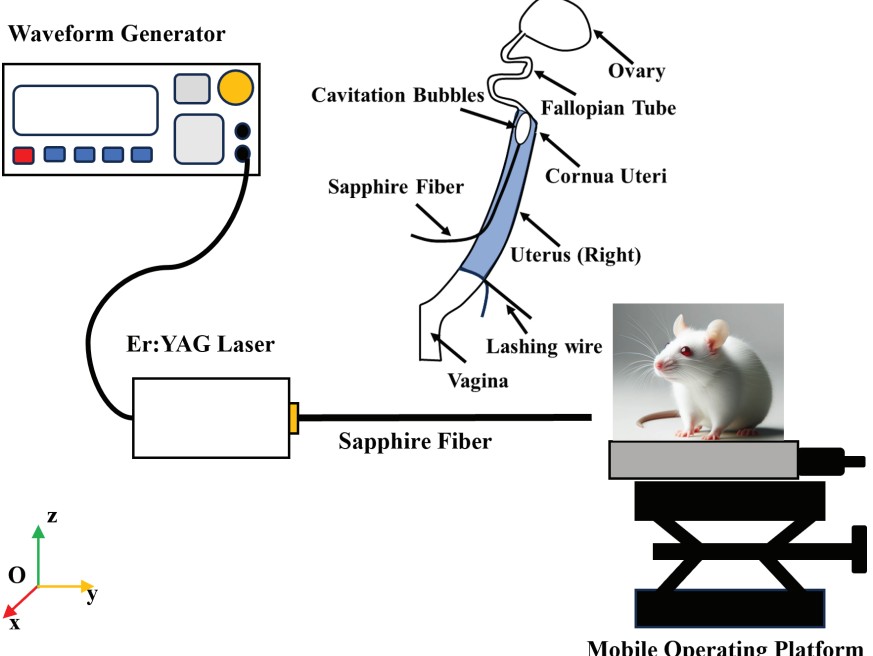

**Fig 3. Schematic diagram of the experimental platform for LICB treatment of FTO.**

distilled water. The goal was to assess the relationship between laser energy and the dimensions of the resulting cavitation bubbles. The experimental arrangement is detailed in Fig 4. The laser system employed for the in vitro experiments was consistent with that used in the experimental platform, ensuring uniformity in the experimental conditions. The experimental liquid, distilled water, was contained in an acrylic tank with a volume of $6 \times 6 \times 6\,\text{cm}^3$ and a depth of 4.5 cm. The fiber's placement was 2 cm beneath the water surface. To mitigate the influence of gravity on bubble size, the laser was oriented to induce cavitation bubbles horizontally in the liquid.

The dynamics of LICB were examined at laser energies of 8 mJ, 12.7 mJ, 15.3 mJ, and 19.7 mJ. A high-speed camera (SA5, Photron, Japan) with a maximum frame rate of 1,000,000 was used to record the process. The laser's activation was controlled by square wave pulses from a waveform generator (DG1002U, Rigol, China), with the trigger signal simultaneously sent to the high-speed camera, commencing the LICB recording process at 100,000 frames per second. Background lighting (10 cm×10 cm, max brightness 7000 lumens) served to differentiate cavitated from non-cavitated areas in the images. For precision, ten cavitation bubbles were sequentially captured at each energy setting.

During the in vitro investigations, illustrated in Fig 5, a correlation was observed between the increasing laser energy and the enlargement of cavitation bubble dimensions, both in maximum length (blue bar) and maximum width (oringe bar). A critical observation was made regarding the $\gamma$ ratio, defined as $\gamma = Rv / W_{\max}$, where $Rv$ is the diameter of the uterine horn and $W_{\max}$ represents the maximum width of the cavitation bubble. A $\gamma$ ratio less than 1 indicates a heightened risk of the cavitation bubbles causing unintended damage to the tubal walls. This finding underscores the need for careful calibration of laser parameters to minimize potential harm [36] .

Measurements of the internal diameters at the uterine horns in two control specimens revealed dimensions of 1.28 mm and 1.35 mm. Based on these measurements and the observed bubble dynamics, laser energies of 8 mJ, 12.7 mJ, and 15.3 mJ were strategically selected for the experimental procedures. These specific laser parameters, chosen to optimize efficacy while minimizing risk, are comprehensively detailed in Table 1. This selection aimed

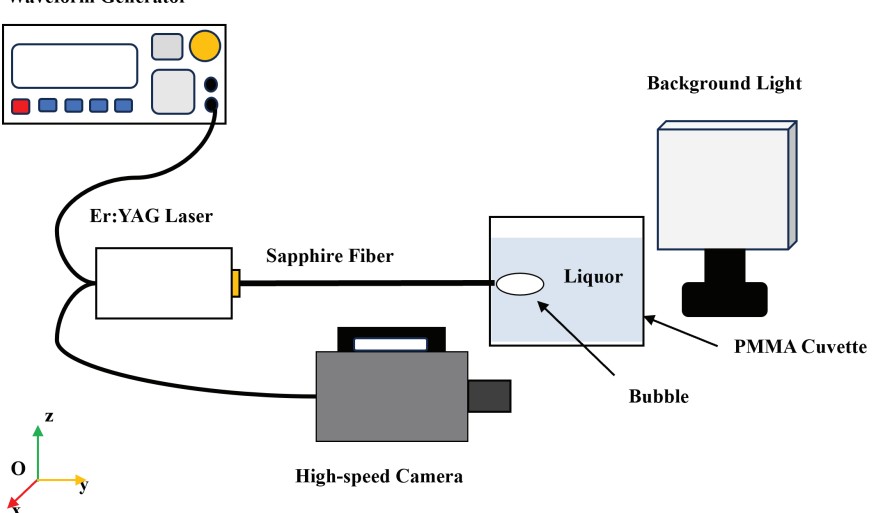

**Fig 4. Schematic diagram of Er:YAG pulse LICB experimental setup.**

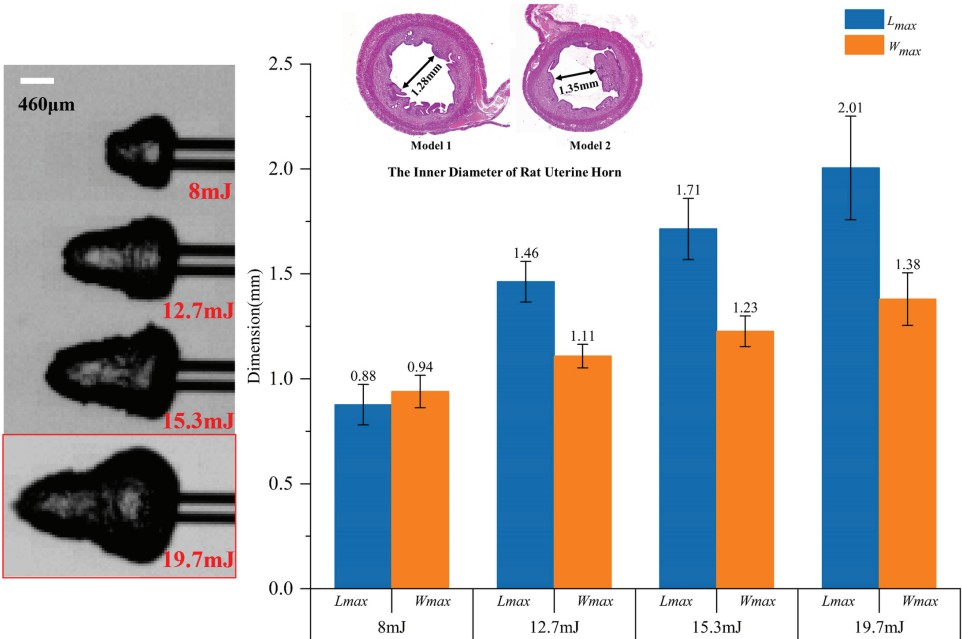

**Fig 5. Shows the maximum size of LICB at different laser energy levels.**

**Table 1. Experimental parameters for LICB treatment of FTO.**

| Pulse width (μs) | Wavelength (μm) | Frequency (Hz) | Energy (mJ) |
|---|---|---|---|
| 200 | 2.94 | 10 | 8 |
| | | | 12.7 |
| | | | 15.3 |

to balance effective fallopian tube recanalization with the safety and integrity of the tubal tissue.

## LICB treatment for animal FTO experiment

In this research, 9 out of the 11 rats in the model group were randomly selected for the Experimental group, while the remaining 2 rats were used as the Model group. Rats assigned to the Experimental Group were randomly allocated into three subgroups, each designated to receive one of three different laser energy levels for recanalization experiments: 8 mJ, 12.7 mJ, and 15.3 mJ. General anesthesia was administered using 10% hydrochloric acid cerazine (5 ml/kg) supplied by Jilin Huamu Animal Health Care Co., Ltd. (veterinary medicine number 070012926). This facilitated the creation of an abdominal incision to expose the Y-shaped uterus-fallopian tube-ovary complex of the rats. The experimental process of LICB Treatment is depicted in Fig 6.

The model's effectiveness was verified by injecting a methylene blue solution in the direction of the uterus-fallopian tube and observing its ability to permeate into the ovaries (Fig 6a). The uterus was consistently filled with this solution to aid in the process. Given that the absorption peak of the 2.94-micron wavelength laser coincides with water molecules, the dynamics of bubble formation were analogous in both methylene blue solution and distilled water. This similarity was critical for the experiment's consistency. A 450μm sapphire optical

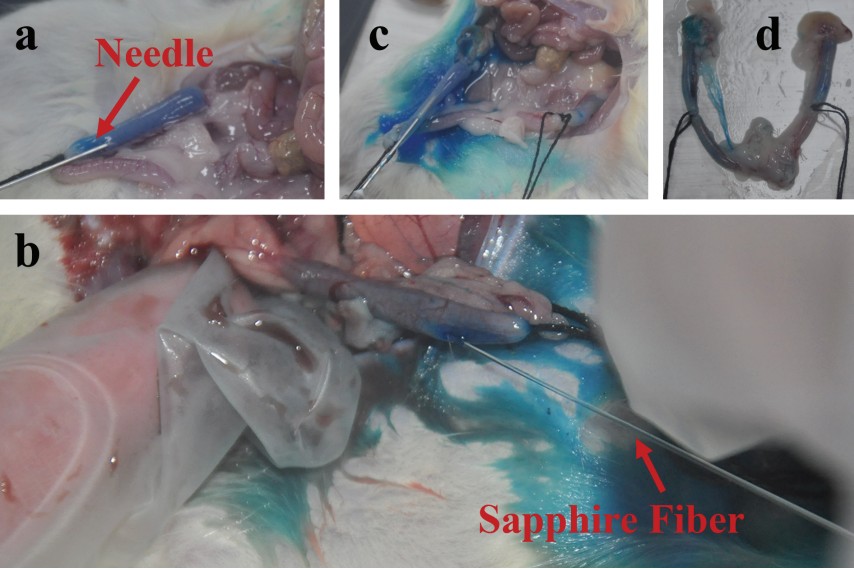

**Fig 6. The process of LICB treatment for FTO.** (a) FTO modeling inspection; (b) LICB treatment experiment; (c) Verification of LICB treatment effectiveness; (d) Comparison of the reproductive systems in rats with and without LICB treatment.

fiber was strategically positioned approximately 1–2 mm into the uterine horn close to the fallopian tube, directing the laser output to the targeted area in the uterus of the rats (Fig 6b).

Each subgroup was subjected to its respective pulsed laser energy (8 mJ, 12.7 mJ, or 15.3 mJ) at a frequency of 10 Hz, aiming to recanalize the fallopian tube obstruction. The treatment exploited the shock jet produced by the cavitation bubbles induced by the laser, with an exposure time set to 10 seconds. The effectiveness of the recanalization process was evaluated post-experiment by assessing the flow of methylene blue solution into the ovarian side, thereby determining the success rate of the recanalization attempts (Fig 6c).

## Evaluation criteria

The effectiveness of the experimental procedure was ascertained through a two-dimensional evaluation encompassing macroscopic inspection of the uterus and fallopian tubes, as well as microscopic assessment of histopathological changes in the fallopian tube tissues. The evaluation protocol was bifurcated into two main components:

1. **Macroscopic Inspection:** This phase entailed a thorough examination of the fallopian tube tissues in each rat group. Key observational parameters included tissue color, morphology, and size. Additionally, any potential laser-induced damages were meticulously identified. In the evaluation of fallopian tube patency, methylene blue solution was administered intraluminally within the uterus. The subsequent traversal of this solution through the fallopian tubes towards the ovaries was meticulously monitored, providing an indicator of tubal patency.

2. **Histopathological Analysis:** After the experimental procedures, fallopian tube tissues were fixed in formalin, followed by a sequence of dehydration steps using a graded ethanol series, then clarified with dimethylformamide, and embedded in paraffin. The paraffin-embedded sections were subsequently dewaxed in water and processed

through a series of solutions: xylene (20 minutes, repeated twice), absolute ethanol (5 minutes, twice), 95% ethanol (1 minute), and 80% ethanol (2 minutes), all followed by a thorough rinse in running water. The sections were stained with Hematoxylin for 5 minutes, subjected to differentiation in differentiation fluid, washed with tap water, and then stained with Eosin for 5 minutes. Following staining, sections were dehydrated, cleared, and mounted for microscopic evaluation to ascertain the degree of damage inflicted on the fallopian tube walls by the LICB.

3. **Scoring Criteria:** The assessment criteria for evaluating the efficacy of LICB treatment on FTO encompass a scoring system, ranging from 0 to 2. The criteria used for this evaluative process are systematically outlined in Table 2.

This scoring framework is meticulously designed to evaluate two key aspects: tubal patency and the extent of damage to uterine and fallopian tube. Each parameter within this system is stratified into different levels, allowing for a nuanced and comprehensive assessment of both the restorative effects of LICB on tubal patency and the potential adverse impacts on the structural integrity of uterine and fallopian tube. These criteria provide a structured and quantifiable approach to gauge the therapeutic outcomes of LICB treatment in alleviating FTO. Each score was independently assessed by three experienced gynecologists.

## Results

### Fallopian tube patency

Upon the conclusion of the experimental trials, the patency of the fallopian tubes was assessed for each of the three laser energy settings, based on the established laser evaluation standards. The specific experimental procedures, laser parameters, and results are shown in Table 3. In this experiment, two rats failed to achieve bilateral fallopian tube obstruction, while two others had successful obstruction on only one side. The failures were characterized by incomplete fallopian tube obstruction, tube abnormalities, and excessively narrow tubes. These issues may be related to potential errors in drug dosing during the modeling process, as well as factors such as fallopian tube infections leading to developmental abnormalities in the rats. The overall success rate of the FTO model was approximately 73%.

In the entirety of efficacious trials, the cumulative success rate of recanalization in rat models afflicted with fallopian tube obstruction, treated via LICB methodology, stood at 75%. The findings, depicted in Fig 7, revealed a gradient of effectiveness across the energy levels. The recanalization rate at 8 mJ was the lowest, registering at only 33.3%. This modest success rate is likely attributable to the insufficiency of the laser energy, which resulted in the generation of less powerful jets and shock waves from the collapsing cavitation bubbles.

In contrast, the application of 12.7 mJ laser energy achieved a considerably higher recanalization rate of 83.3%, suggesting its effectiveness. The instances where recanalization did not

**Table 2. Evaluation criteria for the effectiveness of LICB therapy in treating FTO.**

| Parameter | Description |
|---|---|
| Tubal patency | Score 0: Completely blocked, no dye passage. |
| | Score 1: Partially patent, limited dye passage. |
| | Score 2: Completely patent, unrestricted dye passage. |
| Damage | Score 0: Both visible and histological damage, severe. |
| | Score 1: Visible or histological damage, not severe. |
| | Score 2: No visible or histological damage. |

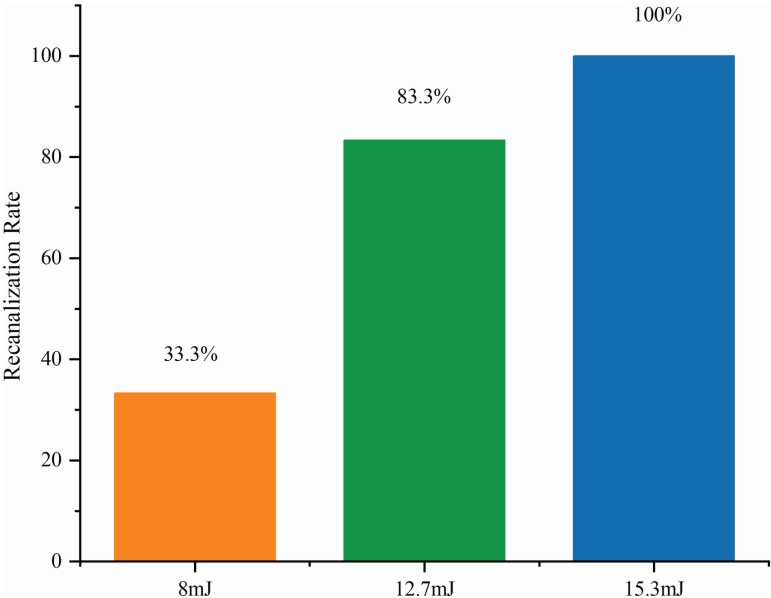

**Fig 7. Recanalization rates of fallopian tubes at different laser energies.**

occur at this energy level could be linked to potential misalignment of the optical fiber with the direction of the FTO during the procedure, leading to the absorption of the bubble energy by the uterine or fallopian tube walls.

Notably, a 100% success rate in recanalization was attained with 15.3 mJ laser energy, underscoring the premise that increased laser energy can significantly improve recanalization outcomes. However, it is imperative to conduct thorough post-experiment examinations of both the external appearance and internal pathology of the fallopian tubes. Such analyses are essential to ascertain and mitigate any potential damage to the uterus or fallopian tube walls that might arise from the application of excessive laser energy.

### Laser-induced tissue damage

Comparative observational analyses were performed on the external morphology of uterine, fallopian tube, and ovarian tissues among the blank, control, and experimental groups, as presented in Fig 8.

1. **Control Group (Fig 8a).** The tissues of the fallopian tubes, ovaries, and uterus in this group appeared healthy and vibrant. They exhibited a pale red hue, indicative of normalcy. The fallopian tubes were wreath-like and closely adhered to the ovaries, suggesting typical anatomical structure. Additionally, the uterus was characterized by its slender diameter and well-developed musculature, aligning with standard physiological attributes.
2. **Model Group (Fig 8b).** A noticeable deviation from the control group was observed in this set. The tissues of the fallopian tubes, ovaries, and uterus were paler in color, suggesting potential pathological changes. Both the fallopian tubes and ovaries exhibited enlargement, and there was significant adhesion observed between them, indicative of pathological alterations. Moreover, the uterus in this group was characterized by a thickened diameter, further suggesting the impact of the induced condition.

**Table 3. Experiment process, laser parameters and recanalization results for the treatment of rat FTO with LICB.**

| Group | No. | Weight (g) | Left/Right (L/R) | Modeling Obstruction (Y/N) | Experimental parameter | Result |
|---|---|---|---|---|---|---|
| 8.3 mJ | 1 | 196.7 | L | N | | |
| | | | R | N | | |
| | 2 | 200 | L | N | | |
| | | | R | Y | Applied laser 12.7 mJ, 10 Hz, for about 10 seconds, snapping sounds without recanalization. | No Recanalization |
| | 3 | 266 | L | Y | Applied laser 12.7 mJ, 10 Hz, for about 10 seconds, Partially recanalization. | Partially Recanalization |
| | | | R | Y | Laser energy 8 mJ, 10 Hz, applied for about 15 seconds, without methylene blue solution entering the ovaries. | No Recanalization |
| 12.7 mJ | 4 | 215.3 | L | Y | 12.7 mJ, 10 Hz, applied for about 15 seconds, recanalization successful | Recanalization |
| | | | R | Y | Applied laser 12.7 mJ, 10 Hz, for about 10 seconds, completely recanalization | Recanalization |
| | 5 | 197 | L | Y | Applied laser 12.7 mJ, 10 Hz, for about 12 seconds, completely recanalization | Recanalization |
| | | | R | Y | Applied laser 12.7 mJ, 10 Hz, for about 12 seconds, completely recanalization | Recanalization |
| | 6 | 270 | L | Y | 12.7 mJ, 10 Hz, for approximately 10 seconds, possibly penetrating the uterine wall (deviation?) | No Recanalization |
| | | | R | Y | Applied laser 12.7 mJ, 10 Hz, for about 10 seconds, completely recanalization | Recanalization |
| 15.3 mJ | 7 | 179.4 | L | N | | |
| | | | R | N | | |
| | 8 | 184.2 | L | Y | Applied laser 15.3 mJ, 10 Hz, for about 10 seconds, noticeable congestion spots, recanalization. | Recanalization |
| | | | R | Y | Applied laser 15.3 mJ, 10 Hz, for about 10 seconds, completely recanalization | Recanalization |
| | 9 | 190.6 | L | Y | Applied laser 15.3 mJ, 10 Hz, for about 8 seconds, noticeable congestion spots, recanalization. | Recanalization |
| | | | R | N | | |

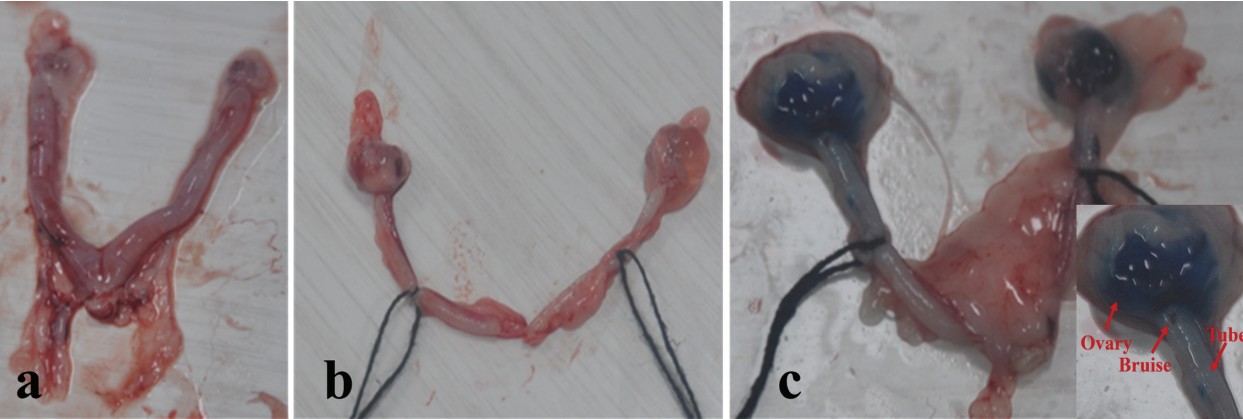

**Fig 8. Images of the rats reproductive system.** a. Control group; b. Model group; c. Experimental group.

3. **Exprimental Group (Fig 8c).** The general appearance of this group bore resemblance to the model group. However, a notable difference was the ability of the Methylene Blue solution to permeate into the ovaries, suggesting some degree of recanalization. The image, representative of the fallopian tubes in the 15.7 mJ laser energy recanalization

subgroup, showed visible hemorrhagic spots on the fallopian tube walls. This observation could be attributed to the application of excessive laser energy, highlighting a potential side effect of the recanalization process at higher energy levels.

Histopathological evaluations were meticulously carried out on the fallopian tube sections from the blank, control, and experimental groups as Illustrated in Fig 9. For each group, two distinct sections of the fallopian tube were examined: one proximal to the uterine horn and the other adjacent to the ovary. This approach enabled a comprehensive comparative analysis.

1. **Control Group:** The histology of the fallopian tube tissue in this group exhibited a well-defined structure. The lumen was patent, and the epithelial cells were neatly arranged, showcasing an abundance of cilia. Notably, there were no significant indications of inflammatory cell infiltration or fibrous tissue proliferation, indicating a healthy state of the tissue.

2. **Model Group:** Contrasting with the control group, the fallopian tube tissue in the model group presented a less distinct structure. The epithelial cells were disorderly arranged, accompanied by a marked presence of inflammatory cells and a pronounced proliferation of fibrous tissue. This pathological change led to the adhesion of ciliary tissue, resulting in the narrowing and eventual occlusion of the lumen, characteristic of the modeled pathological condition.

3. **Experimental Group:** The images displays a HE stained pathological section of the fallopian tube from the 12.7 mJ laser energy recanalization subgroup.In this group, the fallopian tube tissue demonstrated a clear structural composition with an enlarged lumen. The previously proliferated fibrous tissue appeared disrupted, and the ciliary tissue was distinctly separated with minimal shedding. This observation indicated a successful restoration of lumen patency, aligning with the objectives of the experimental treatment.

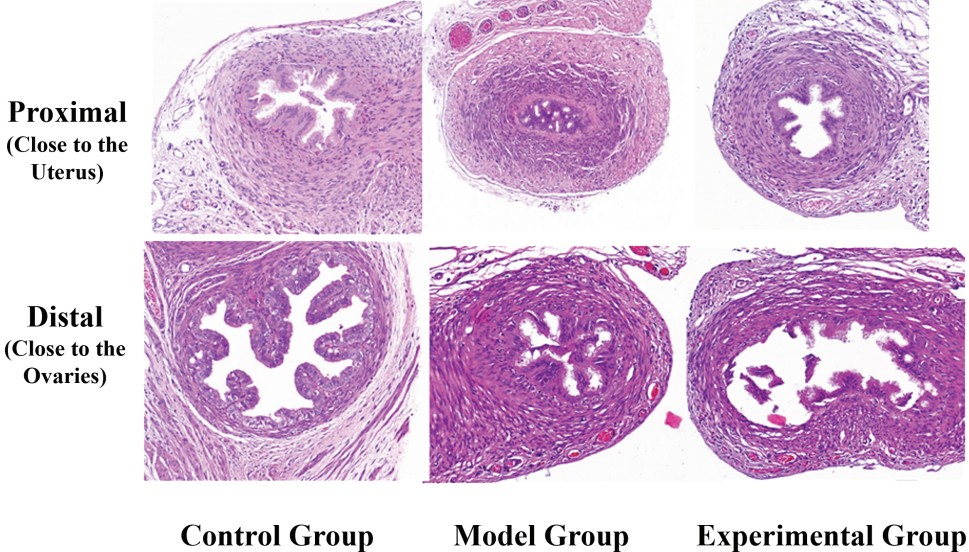

**Proximal** (Close to the Uterus)

**Distal** (Close to the Ovaries)

**Control Group**          **Model Group**          **Experimental Group**

**Fig 9. Pathological sections of the control group, model group, and experimental group.**

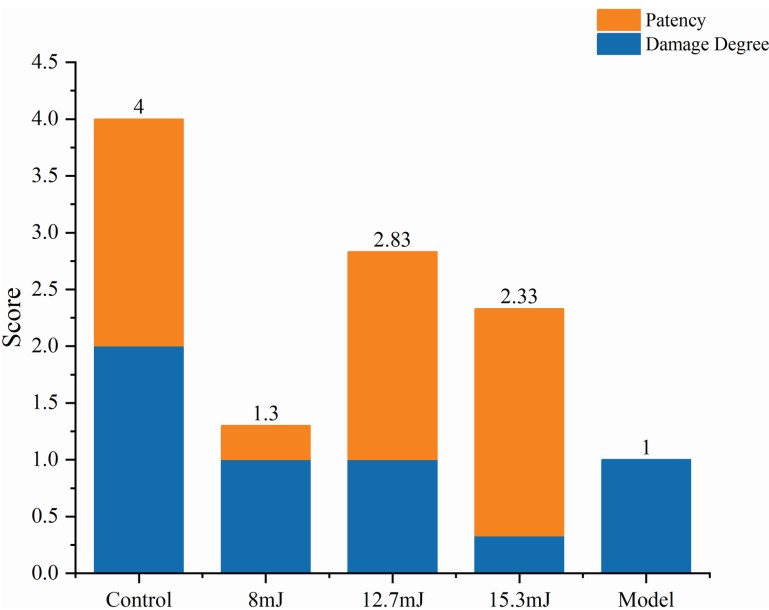

**Fig 10. Scores of fallopian tubes in different groups based on evaluation criteria.**

## Recanalization score

Based on the evaluation of appearance, histopathology, and patency assessed through methylene blue solution injection, a comprehensive assessment was conducted on the uterus and fallopian tubes of the experimental groups treated with varying laser energies, as well as the model and control groups. The evaluation encompassed two dimensions for each group, as detailed in Table 2, with the scoring outcomes depicted in Fig 10. The graph's blue columns score fallopian tube damage, while orange columns rate patency. Combined, they indicate overall experimental efficacy.

In the control group, comprising healthy mice, no apparent damage was observed either in appearance or histopathologically, and the fallopian tubes exhibited clear patency. Contrarily, in the model group, inflammation led to fallopian tube obstruction, as evidenced by microscopic observations of damaged and adherent ciliary structures.

Within the experimental groups, varying degrees of success were observed. The group treated with 8 mJ laser energy scored the lowest, primarily attributed to the insufficient laser power, which resulted in suboptimal recanalization rates and ineffective removal of fibrous tissue in the fallopian tubes. Conversely, the 15.3 mJ group, despite achieving the highest recanalization rate, presented noticeable burn marks on some uteri and fallopian tubes post-treatment. This issue could be linked to the high power of the 15.3 mJ laser, where the Moses effect of the bubbles likely caused direct laser irradiation on the tubal or uterine walls, damaging the internal structures.

Notably, the 12.7 mJ group emerged as the most effective, registering the highest overall score. This subgroup accomplished a recanalization rate of 83.3% without inducing significant additional damage to the fallopian tube walls, thus identifying 12.7 mJ as the most optimal laser energy parameter for this procedure.

## Discussion

The fallopian tubes, integral to the reproductive process due to their role in transporting sperm and eggs and as the site of fertilization, are often implicated in infertility cases [37]. Fallopian tube pathology, especially when it affects tubal patency, is a common cause of female infertility [38]. Damage to the fallopian tubes can arise from various conditions, including inflammation, endometriosis, and surgical trauma, with inflammatory causes being notably prevalent [39,40]. These often lead to proximal adhesion and distal blockage characterized by pus and fluid accumulation.

This study explored the potential of LICB technology in a rat model of scar-induced FTO. The model, established by injecting phenol paste into the fallopian tubes, created a scar-based obstruction, which was then targeted using an Er:YAG laser on a custom-built laser surgery platform. The procedure harnessed the power of shockwave jets from LICB, and the optimal laser parameters were determined through extensive in vitro experiments and calculations.

The experimental results highlighted the efficacy variability across different laser energies. While the lowest energy (8 mJ) yielded a modest recanalization rate, the highest energy (15.3 mJ) resulted in complete recanalization but also caused significant thermal damage. The intermediate energy (12.7 mJ) emerged as the most effective, balancing high recanalization rates with minimal tubal damage.

Postoperative histopathological analyses revealed restored structural integrity and luminal patency of the fallopian tubes, with minimal signs of inflammation, fibrous tissue proliferation, or mucosal damage. This finding, coupled with the minimal hemorrhage observed during the procedure, underscores the potential of LICB as a viable treatment method.

## Conclusion

The experimental results substantiate the efficacy of LICB technology in effectively clearing FTO while causing acceptable levels of damage. This underscores its potential as a valuable therapeutic approach, meriting further exploration to enhance tubal recanalization. However, as this study represents an initial phase of research, limitations are acknowledged, including the use of the minimum sample size with statistical significance to reduce animal usage, a relatively simplistic modeling approach and the differences in sample sizes across subgroups for the variability in FTO rates. Despite these constraints, the findings offer encouraging insights for the treatment of FTO. Future work will aim to refine the experimental protocol, such as conducting post-recanalization observations to ascertain the re-obstruction rate and the effective pregnancy rate post-treatment, which will be key focuses of ongoing research.

## Acknowledgments

The symbol '‡' next to the author's name in the upper right corner represents co-first authorship.

## Author contributions

**Conceptualization:** Yong Bi.

**Data curation:** Dongdong Wang, Xuehong Zhu, Weinan Gao.

**Formal analysis:** Zhong Lin, Weinan Gao.

**Investigation:** Dongdong Wang, Yong Bi.

**Methodology:** Dongdong Wang.

**Project administration:** Fang Yang, Xuehong Zhu.

**Resources:** Fang Yang.

**Supervision:** Yong Bi.

**Validation:** Zhong Lin.

**Writing – original draft:** Dongdong Wang.

**Writing – review & editing:** Dongdong Wang.

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
