## [Decision Letter · Decision Letter 0]

14 Nov 2024

PONE-D-24-05365Exploratory Study on the Application of Laser-Induced Cavitation Bubble in the Treatment of Fallopian Tube ObstructionPLOS ONE

Dear Dr. Wang,

Thank you for submitting your manuscript to PLOS ONE. After careful consideration, we feel that it has merit but does not fully meet PLOS ONE’s publication criteria as it currently stands. Therefore, we invite you to submit a revised version of the manuscript that addresses the points raised during the review process.

We look forward to receiving your revised manuscript.

Kind regards,

Patrick Ifeanyi Okonta, MBBCh, MPH, FWACS, FMCOG, MD, DRH

Academic Editor

PLOS ONE

Journal Requirements:

4. We note that your Data Availability Statement is currently as follows: “All relevant data are within the manuscript and in Supporting Information files.”

Please confirm at this time whether or not your submission contains all raw data required to replicate the results of your study. Authors must share the “minimal data set” for their submission. PLOS defines the minimal data set to consist of the data required to replicate all study findings reported in the article, as well as related metadata and methods (https://journals.plos.org/plosone/s/data-availability#loc-minimal-data-set-definition). For example, authors should submit the following data: - The values behind the means, standard deviations and other measures reported; - The values used to build graphs; - The points extracted from images for analysis. Authors do not need to submit their entire data set if only a portion of the data was used in the reported study. If your submission does not contain these data, please either upload them as Supporting Information files or deposit them to a stable, public repository and provide us with the relevant URLs, DOIs, or accession numbers. For a list of recommended repositories, please see https://journals.plos.org/plosone/s/recommended-repositories. If there are ethical or legal restrictions on sharing a de-identified data set, please explain them in detail (e.g., data contain potentially sensitive information, data are owned by a third-party organization, etc.) and who has imposed them (e.g., an ethics committee). Please also provide contact information for a data access committee, ethics committee, or other institutional body to which data requests may be sent. If data are owned by a third party, please indicate how others may request data access.

5. Please note that funding information should not appear in any section or other areas of your manuscript. We will only publish funding information present in the Funding Statement section of the online submission form. Please remove any funding-related text from the manuscript.

Reviewers' comments:

Reviewer's Responses to Questions

**Comments to the Author**

1. Is the manuscript technically sound, and do the data support the conclusions?

Reviewer #1: Yes

Reviewer #2: Yes

2. Has the statistical analysis been performed appropriately and rigorously? 

Reviewer #1: N/A

Reviewer #2: No

3. Have the authors made all data underlying the findings in their manuscript fully available?

Reviewer #1: Yes

Reviewer #2: Yes

4. Is the manuscript presented in an intelligible fashion and written in standard English?

Reviewer #1: Yes

Reviewer #2: Yes

5. Review Comments to the Author

Reviewer #1: This is a well-designed and valuable experimental study. I've got some questions and suggestions.

1. It seems that there should be 11 mice in the experimental group, but Table 3 shows the results of 9 mice. This difference is not mentioned in the article.

2. Were there mice that failed to form FTOs (we understand from Table 3 that there were, but this is not mentioned in the text)?

3. The mice in the subgroups in the study group did not show similar rates of FTO. Therefore, different numbers of applications were made in each of the three subgroups in the evaluation of different energy levels of laser. Is it possible to statistically evaluate the results after the application here? This is not mentioned in the text.

4. The quality of some figures needs to be improved.

5. HE was always used as an abbreviation in the text, it must be clearly written on the first use.

Reviewer #2: I accepted the work largely because it is a preliminary attempt targeted at resolving tubal patency related infertility. This is important because it might serve as a check on the rising cost of In Vitro fertilisation (IVF). Efforts like this one should therefore be encouraged. The limitations were clearly acknowledged by the authors; "this study represents an initial phase of research, limitations are acknowledged, including the use of the minimum sample size with statistical significance to reduce animal usage, and a relatively simplistic modeling approach." The paper is novel. This should encourage more efforts in the area and ultimately lead to the desired effect of reduction in IVF cost.

6. PLOS authors have the option to publish the peer review history of their article (what does this mean?). If published, this will include your full peer review and any attached files.

Reviewer #1: No

Reviewer #2: **Yes: **KINGSLEY AGHOLOR

---

## [Author Response · Author response to Decision Letter 1]

16 Dec 2024

Response to Reviewers

Dear editors:

We would like to sincerely thank you and the reviewers for the time and effort invested in reviewing our manuscript titled "Exploratory Study on the Application of Laser-Induced Cavitation Bubble in the Treatment of Fallopian Tube Obstruction" (Manuscript ID: [PONE-D-24-05365R1]). We greatly appreciate the constructive feedback provided, which has helped us improve the quality of our work. We have carefully addressed all comments and suggestions, and we believe the revisions have enhanced the manuscript. Below, we provide a detailed point-by-point response to each comments of the academic editor and reviewer(s).

1) Responds to point of the academic editor

Q1. Please ensure that your manuscript meets PLOS ONE's style requirements, including those for file naming. The PLOS ONE style templates can be found at https://journals.plos.org/plosone/s/file?id=wjVg/PLOSOne_formatting_sample_main_body.pdf and https://journals.plos.org/plosone/s/file?id=ba62/PLOSOne_formatting_sample_title_authors_affiliations.pdf

R:Thank you for your feedback. In response to your request, we have revised the manuscript using the LaTeX template provided by PLOS ONE. We have made every effort to ensure that the formatting adheres to the journal’s style requirements. Both the revised and clean versions of the manuscript are now provided for your review.We appreciate your consideration and are happy to make any further adjustments if needed.

Q2. To comply with PLOS ONE submissions requirements, in your Methods section, please provide additional information regarding the experiments involving animals and ensure you have included details on (1) methods of sacrifice, (2) methods of anesthesia and/or analgesia, and (3) efforts to alleviate suffering.

R: We have carefully reviewed the Methods section and made the necessary revisions. We have added detailed information in line 140 regarding the methods of euthanasia, ensuring compliance with AVMA guidelines. Additionally, we have provided specific details on the anesthesia and analgesia methods used, as well as the efforts made to minimize animal suffering during surgery. These changes have been reflected in the revised manuscript line 120 and line 160. We greatly appreciate your suggestions, which have significantly improved the quality of our paper.

Q3.We note that the grant information you provided in the ‘Funding Information’ and ‘Financial Disclosure’ sections do not match. When you resubmit, please ensure that you provide the correct grant numbers for the awards you received for your study in the ‘Funding Information’ section.

R:Thank you for pointing out the discrepancy. We have revised the f Financial Disclosure as the following statement: “The work has been funded by Nanning Scientific Research and Technological Development Plans Project (No.20233067) and National Natural Science Foundation of China (No.62205350). The funders had no role in study design, data collection and analysis, decision to publish, or preparation of the manuscript.”

As we are currently unable to make the changes directly to the Financial Disclosure section, we have attached this as a separate "Other" file. Please let us know if any further information is required. Thank you for your understanding.

Q4. We note that your Data Availability Statement is currently as follows: “All relevant data are within the manuscript and in Supporting Information files.”

Please confirm at this time whether or not your submission contains all raw data required to replicate the results of your study. Authors must share the “minimal data set” for their submission. PLOS defines the minimal data set to consist of the data required to replicate all study findings reported in the article, as well as related metadata and methods (https://journals.plos.org/plosone/s/data-availability#loc-minimal-data-set-definition). For example, authors should submit the following data: - The values behind the means, standard deviations and other measures reported; - The values used to build graphs; - The points extracted from images for analysis. Authors do not need to submit their entire data set if only a portion of the data was used in the reported study. If your submission does not contain these data, please either upload them as Supporting Information files or deposit them to a stable, public repository and provide us with the relevant URLs, DOIs, or accession numbers. For a list of recommended repositories, please see https://journals.plos.org/plosone/s/recommended-repositories. If there are ethical or legal restrictions on sharing a de-identified data set, please explain them in detail (e.g., data contain potentially sensitive information, data are owned by a third-party organization, etc.) and who has imposed them (e.g., an ethics committee). Please also provide contact information for a data access committee, ethics committee, or other institutional body to which data requests may be sent. If data are owned by a third party, please indicate how others may request data access.

R: We appreciate your feedback and are happy to confirm our commitment to sharing all data necessary to replicate the results of our study. The data for Figures 5 and 9 have been uploaded to a public repository for easy access and sharing. The download URL is: https://doi.org/10.6084/m9.figshare.27936726. Additionally, the data for Figure 7 has been disclosed in the supplementary files provided with the submission.We hope this fulfills the journal’s requirements for data availability. Please let us know if further clarification or additional steps are needed. Should you require any further information, please do not hesitate to contact us.

Q5. Please note that funding information should not appear in any section or other areas of your manuscript. We will only publish funding information present in the Funding Statement section of the online submission form. Please remove any funding-related text from the manuscript.

R:Thank you for the clarification. We will ensure that all funding-related information is removed from the manuscript text and included only in the Funding Statement section of the online submission form as per your requirements. We appreciate your guidance.

Q6. Please review your reference list to ensure that it is complete and correct. If you have cited papers that have been retracted, please include the rationale for doing so in the manuscript text, or remove these references and replace them with relevant current references. Any changes to the reference list should be mentioned in the rebuttal letter that accompanies your revised manuscript. If you need to cite a retracted article, indicate the article’s retracted status in the References list and also include a citation and full reference for the retraction notice.

R:Thank you for pointing this out. We will carefully review the reference list to ensure it is complete and accurate. Any retracted articles will either be replaced with current, relevant references or cited with their retracted status clearly indicated, along with the full reference for the retraction notice. All changes to the reference list will be detailed in the rebuttal letter. We appreciate your guidance.

2) Responds to point of the reviewers

Reviewer #1: This is a well-designed and valuable experimental study. I've got some questions and suggestions.

Q1:It seems that there should be 11 mice in the experimental group, but Table 3 shows the results of 9 mice. This difference is not mentioned in the article.

R:Thank you for your insightful comment. To clarify, we created a total of 11 rat models, of which 2 were assigned to the Model Group for comparison with the experimental results, while the remaining 9 rats were assigned to the Experimental Group. Therefore, the results shown in Table 3 represent data from the 9 rats in the Experimental Group. We have also provided this clarification in line 198 of the manuscript. We hope this resolves the discrepancy, and we appreciate your careful review.

Q2:Were there mice that failed to form FTOs (we understand from Table 3 that there were, but this is not mentioned in the text)?

R:Thank you for your comment. We have now added the information regarding the number of mice that failed to form FTOs, as well as the success rate of modeling, in line 263 of the manuscript. We hope this clarifies the issue. Thank you for your insightful feedback.

Q3:The mice in the subgroups in the study group did not show similar rates of FTO. Therefore, different numbers of applications were made in each of the three subgroups in the evaluation of different energy levels of laser. Is it possible to statistically evaluate the results after the application here? This is not mentioned in the text.

R:Thank you for your thoughtful comment. As you noted, while our animal experiments yielded promising results, the smaller sample size in the Experimental Group and the variability in FTO rates led to differences in subgroup sample sizes, which may have increased the uncertainty of our conclusions. We will emphasize this limitation in the conclusion of the manuscript and highlight the need for future studies with larger sample sizes to validate our findings. We appreciate your valuable feedback.

Q4:The quality of some figures needs to be improved.

R:Thank you for your constructive comment. We have thoroughly reviewed all the figures and made efforts to improve their clarity, ensuring that all images now meet the journal's requirements. Additionally, we have adjusted the font sizes in Figures 2, 6, 8, and 9 to improve the readability of the labels and annotations. We hope these changes address your concerns.

Q5:HE was always used as an abbreviation in the text, it must be clearly written on the first use.

R:Thank you for pointing this out. We have now included the full form of "HE" upon its first use in the manuscript line 25, as per your suggestion. We appreciate your careful attention to detail.

Reviewer #2: I accepted the work largely because it is a preliminary attempt targeted at resolving tubal patency related infertility. This is important because it might serve as a check on the rising cost of In Vitro fertilisation (IVF). Efforts like this one should therefore be encouraged. The limitations were clearly acknowledged by the authors; "this study represents an initial phase of research, limitations are acknowledged, including the use of the minimum sample size with statistical significance to reduce animal usage, and a relatively simplistic modeling approach." The paper is novel. This should encourage more efforts in the area and ultimately lead to the desired effect of reduction in IVF cost.

R :We sincerely thank Reviewer #2 for their positive and encouraging feedback. We deeply appreciate your recognition of the importance of our work in addressing tubal patency-related infertility and the potential impact on reducing IVF costs. Your support motivates us to continue our efforts in this area, and we are committed to conducting further in-depth research to address the limitations identified in this preliminary phase. We are grateful for your constructive comments and will strive to build upon this initial work to meet the high expectations you have outlined.

Once again, we would like to express our gratitude to you and the reviewers for your thoughtful suggestions. We are confident that the revisions have strengthened the manuscript, and we look forward to your feedback. Please do not hesitate to contact us if further clarification is required.

Sincerely,

Wang Dongdong

---

## [Editor Report · Decision Letter 1]

20 Dec 2024

Exploratory Study on the Application of Laser-Induced Cavitation Bubble in the Treatment of Fallopian Tube Obstruction

PONE-D-24-05365R1

Dear Dr. Bi,

We’re pleased to inform you that your manuscript has been judged scientifically suitable for publication and will be formally accepted for publication once it meets all outstanding technical requirements.

Kind regards,

Patrick Ifeanyi Okonta, MBBCh, MPH, FWACS, FMCOG, MD, DRH

Academic Editor

PLOS ONE
---

## [Editor Report · Acceptance letter]

PONE-D-24-05365R1

PLOS ONE

Dear Dr. Bi,

I'm pleased to inform you that your manuscript has been deemed suitable for publication in PLOS ONE. Congratulations! Your manuscript is now being handed over to our production team.

Kind regards,

on behalf of

Professor Patrick Ifeanyi Okonta

Academic Editor

PLOS ONE